# A Radiomics Approach to Predict the Emergence of New Hepatocellular Carcinoma in Computed Tomography for High-Risk Patients with Liver Cirrhosis

**DOI:** 10.3390/diagnostics11091650

**Published:** 2021-09-09

**Authors:** Eric Tietz, Daniel Truhn, Gustav Müller-Franzes, Marie-Luise Berres, Karim Hamesch, Sven Arke Lang, Christiane Katharina Kuhl, Philipp Bruners, Maximilian Schulze-Hagen

**Affiliations:** 1Department of Diagnostic and Interventional Radiology, University Hospital, RWTH Aachen, Pauwelsstrasse 30, 52074 Aachen, Germany; dtruhn@ukaachen.de (D.T.); gumueller@ukaachen.de (G.M.-F.); ckuhl@ukaachen.de (C.K.K.); pbruners@ukaachen.de (P.B.); 2Department of Internal Medicine III, University Hospital, RWTH Aachen, Pauwelsstrasse 30, 52074 Aachen, Germany; mberres@ukaachen.de (M.-L.B.); khamesch@ukaachen.de (K.H.); 3Center for Integrated Oncology Aachen Bonn Cologne Duesseldorf (CIO ABCD), University Hospital, RWTH Aachen, Pauwelsstrasse 30, 52074 Aachen, Germany; 4Department of Surgery and Transplantation, University Hospital, RWTH Aachen, Pauwelsstrasse 30, 52074 Aachen, Germany; svlang@ukaachen.de

**Keywords:** CT, liver cirrhosis, hepatocellular carcinoma, radiomics, tumor prediction

## Abstract

Liver cirrhosis poses a major risk for the development of hepatocellular carcinoma (HCC). This retrospective study investigated to what extent radiomic features allow the prediction of emerging HCC in patients with cirrhosis in contrast-enhanced computed tomography (CECT). A total of 51 patients with liver cirrhosis and newly detected HCC lesions (*n* = 82) during follow-up (FU-CT) after local tumor therapy were included. These lesions were not to have been detected by the radiologist in the chronologically prior CECT (PRE-CT). For training purposes, segmentations of 22 patients with liver cirrhosis but without HCC-recurrence were added. A total of 186 areas (82 HCCs and 104 cirrhotic liver areas without HCC) were analyzed. Using univariate analysis, four independent features were identified, and a multivariate logistic regression model was trained to classify the outlined regions as “HCC probable” or “HCC improbable”. In total, 60/82 (73%) of segmentations with later detected HCC and 84/104 (81%) segmentations without HCC were classified correctly (AUC of 81%, 95% CI 74–87%), yielding a sensitivity of 72% (95% CI 57–83%) and a specificity of 86% (95% CI 76–96%). In conclusion, the model predicted the occurrence of new HCCs within segmented areas with an acceptable sensitivity and specificity in cirrhotic liver tissue in CECT.

## 1. Introduction

Hepatocellular cancer (HCC) is the second most lethal tumor, the sixth most diagnosed cancer worldwide [1], and the primary malignant liver tumor in liver cirrhosis [2]. Each year approximately 1–8% of patients with liver cirrhosis and no signs of liver failure develop HCC [3], corresponding to an approximate 30% risk of developing HCC during one’s lifetime in the presence of cirrhosis [4].

Due to the high risk of HCC development in patients with liver cirrhosis, screening is recommended at regular intervals [3,5]. In a meta-analysis, surveillance has been proven to increase the detection rate of subclinical early-stage HCC and therefore facilitate curative treatment compared with no surveillance [6]. The Barcelona Clinic Liver Cancer (BCLC) staging system is the most used classification with first-line treatment recommendations [7,8]. Ideally, for all treatment options to be applicable, a solitary lesion is detected at a very early stage with <2 cm in size.

The imaging modality of choice for surveillance is contrast-enhanced computed tomography (CECT) followed by magnetic resonance imaging and ultrasound [9,10,11]. According to the Liver Imaging Reporting and Data System (LI-RADS), in cirrhotic patients, a lesion is categorized as definitely HCC if it demonstrates non-rim hyperenhancement in the arterial phase (AP) and is either >2 cm or 1–2 cm and additionally shows ≥50 percent increase in size in ≤6 months or washout appearance in portal vein phase (PVP) [12]. Human performance is limited with lesion-based sensitivity in patients with liver cirrhosis for CECT approximately amounting to 60% and decreasing to 34% for HCC 1.0–1.9 cm in size [13]. One way to improve this is to perform liver magnetic resonance imaging (MRI) scans. For all tumor sizes, comparisons showed significantly better performances for MRI in liver cirrhosis with a sensitivity of approximately 82% [14]. However, the availability of MRI examinations varies locoregionally and not all patients are suitable for this complex examination. This underlines a clinical need for improved detection of small HCC lesions in CECT imaging.

To improve image-based tumor detection, quantitative image descriptors are a promising approach. This research area is usually referred to as radiomics and aims to quantify the morphological appearance of radiological features using specialized computer algorithms [15,16,17]. Recent studies using radiomics in patients with HCC have shown its potential for diagnosis and prognosis [18]. However, no present study has utilized radiomics to not only detect but also predict the development of newly emerging HCCs in patients with liver cirrhosis using CECT imaging. The aim of this study was to establish and validate a set of CECT-based radiomic features to predict de novo development of HCC in outlined regions of cirrhosis.

## 2. Materials and Methods

### 2.1. Study Design and Sample

For this retrospective study, ethical approval was obtained (EK 379-20) and the requirement for informed consent was waived. The study was conducted in accordance with contemporary data protection laws. The entire cohort dataset was acquired from January 2009 to January 2021 records of the institutional picture archiving and communication system (IntelliSpace PACS; Philips, Best, the Netherlands) using a standardized query for patients with liver cirrhosis and a history of HCC who underwent a multiphase CECT surveillance after local tumor therapy (tumor ablation or transarterial tumor embolization). A radiologist with 4 years of experience in abdominal imaging screened for patients who developed new HCC during the surveillance period, as shown in Figure 1. According to the LI-RADS criteria these were defined as newly demarcated lesions with arterial enhancement and late venous washout and a lesion size >1 cm. A few newly detected lesions did not yet meet the formal criteria of definite HCC due to a lesion size <1 cm but appeared otherwise typical for HCC. [12]. These lesions were included if verifiable as HCCs based on further tumor progression in the subsequent follow-up CTs. The presence of HCC in other segments of the liver was not an exclusion criterion. To simplify further denomination, CTs with newly definable HCCs will be referred to as “follow-up CT” (FU-CT) in the following, the chronologically immediate prior examinations will henceforth be referred to as “previous CT” (PRE-CT).

Exclusion criteria were as follows: (a) patients with a period ≥ 300 days between PRE-CT and FU-CT, (b) patients with atypical image features for HCC lesions according to LI-RADS, (c) patients with retrospectively detectable HCC lesion in PRE-CT, (d) CTs with a slice thickness >3 mm and (e) CTs with inadequate arterial contrast phase. If multiple de novo HCCs were detected on FU-CT, a maximum of 3 lesions per CECT were included in the study. As negative controls, additional 22 patients with liver cirrhosis who did not develop new HCC under surveillance were enrolled and liver regions were segmented with respect to an even distribution over all liver lobes and a comparable size to median tumor diameter (MTD) of the case group. For each patient, characteristics such as age, sex, alpha-fetoprotein (AFP) level, MTD, and tumor location, if applicable, were determined.

### 2.2. CT Parameters

All CECT were performed with 128-row spiral CT scanners (Somatom Definition Flash or Somatom Definition AS, Siemens Medical Systems, Erlangen, Germany). The scans were acquired in craniocaudal direction during a single breath-hold with a tube voltage of 120 kV, automated tube current modulation to a quality reference of 240 mAs [19], 0.5 s rotation time, 1 mm or 2 mm slice thickness, and an increment of 0.7 mm. A 1.5 mL/kg body weight bolus of iopromide 370 mg/mL (Ultravist, Bayer, Leverkusen, Germany) was injected intravenously by a power injector. AP was obtained 6 s after automated detection of contrast medium in the aorta at the level of the coeliac trunk with a threshold of 140 Hounsfield units (HU). The protocol requirements used for CECT imaging met the criteria recommended by the LI-RADS guideline [20].

### 2.3. Image Segmentation

Three-dimensional manual segmentation was performed by a radiologist with 4 years’ experience in abdominal imaging, using ITK-SNAP version 3.6 (http://www.itksnap.org, accessed on 2 November 2020). Segmentations were drawn slice-by-slice around the visible borders of the 82 newly emerged HCC lesions in the AP of FU-CT. Next, the corresponding 82 regions of liver tissue were identified in the PRE-CT in AP by the radiologist and matched using anatomical landmarks. For each lesion, manual segmentation was drawn in a similar fashion as for the FU-CT. As intra-individual negative controls, 82 segmentations of the same size were placed in HCC-free liver parenchyma in PRE-CT and FU-CT for each patient, again in such a fashion, that the segmentations covered the same region in both CTs. Segmentations of a not yet evident (PRE-CT) and later newly determined HCC (FU-CT), as well as control regions with tumor-free liver tissue, are illustrated in Figure 2. Additionally, a segmentation was drawn in each of the 22 CTs of patients without HCC-recurrence. All segmentations were validated by a senior radiologist with 8 years of experience in liver CT examinations.

### 2.4. Radiomic Features

Radiomic features were extracted using the PyRadiomics framework [21] and were based on feature definitions as described by the Imaging Biomarker Standardization Initiative (IBSI) [22]. They include first-order statistic features, shape-based features, and texture features (gray level co-occurrence matrix, gray level run length matrix, gray level size zone matrix, neighboring gray-tone difference matrix, and gray level dependence matrix). A total of 105 radiomic features were extracted from each segmentation. To evaluate which radiomic features differentiate between PRE-CT scans with and without HCC, the feature distributions were compared using univariate logistic regression. Features with the highest discriminability between HCC and no HCC (smallest *p*-value) were selected as most promising.

### 2.5. Development of a Prediction Model

Employing the four most promising radiomic features from the univariate analysis, a multivariate logistic regression model from Scikit-learn [23] was used to predict the development of HCC in each segmentation. Each feature was adjusted for mean and standard deviation to achieve uniform scaling between features. Based on the estimated HCC probabilities and the actual occurrence of HCC, the optimal threshold in terms of maximum sensitivity and specificity was determined in a ROC curve. This threshold was then used to classify PRE-CT images into “HCC probable” and “HCC improbable”. Features were tested for multicollinearity using the Spearman correlation coefficient. To further increase the predictive power, a trial was carried out in which patients’ AFP level at the time of the PRE-CT was included in the multivariate logistic regression model to investigate whether additional non-imaging information can increase the accuracy. To account for the scarcity of data, patient-by-patient leave-one-out cross-validation (LOOCV) was used. That is, multiple models were trained by leaving one patient out of the training set while training with the remaining set of patients. This process was repeated until every patient had been in the test set.

### 2.6. Statistical Analysis

All the statistical analyses were performed using python packages scipy and numpy. Confusion matrices and ROC analysis were calculated for each model. *p*-values were calculated using bootstrapping for the ROC analysis and t-tests for the difference in the distribution in univariate analyses. Significance thresholds were adjusted using Bonferroni correction with a baseline significance threshold of 0.05.

## 3. Results

In this study, 85 patients with liver cirrhosis and newly developed HCC during CECT surveillance were screened for eligibility. Thirty-four patients were excluded due to a period ≥300 days between PRE-CT and FU-CT (*n* = 19), atypical image features according to LI-RADS (*n* = 7), retrospectively detectable HCC lesion in PRE-CT (*n* = 5), CTs with a slice thickness >3 mm (*n* = 2) and inadequate arterial contrast phases (*n* = 1). The final group of patients with HCC was composed of 51 patients with a mean age of 69 years and with an interquartile range (IQR) of 63–76 years. 12/51 (24%) patients were female. Within this group, 82 new HCC lesions were observed with a median tumor diameter (MTD) of 1.2 cm and an even distribution between the right and left hepatic lobe. The median time interval between PRE-CT and FU-CT within the interim newly detectable HCC was 121 days (IQR 94–165). For a wider selection and to avoid potential bias, segmentations of 22 patients with liver cirrhosis without HCC-recurrence under surveillance were included in this study. This group had a mean age of 55 years with an interquartile range (IQR) of 51–63 years and 8 (36%) were female. The epidemiologic and clinical characteristics are summarized in Table 1. A total of 186 areas (82 HCCs and 104 liver areas without HCCs) were used for the analysis, yielding a total of 350 segmentations: 82 HCCs and 82 intra-individual control lesions of tumor-free liver tissue, each segmented in PRE-CT and FU-CT and 22 additional segmentations of the control-group (82 × 4 + 22 = 350 segmentations).

Using univariate analysis, four features were identified after Bonferroni correction that allowed differentiation of cirrhotic liver tissue with and without subsequent HCC. Three of them—correlation, informational measure of correlation 1 (IMC1), and informational measure of correlation 2 (IMC2)—are texture features and belong to the class of gray level co-occurrence matrix (GLCM) features [24]. IMC1 and IMC2 both quantify the complexity of the texture, whereas correlation is a measure of the linear dependency of gray level values to their respective voxels in the GLCM. The fourth feature—kurtosis, is a first-order feature and specifies the peaked-ness of the distribution of values in the segmentation. The predictive scores including sensitivity and specificity for all univariate models are given in Table 2. The area under curve (AUC) specified with 95 percent confidence interval (CI) was highest for IMC2 (78%, 95% CI 72–85%), followed by correlation (76%, 95% CI 68–83%), IMC1 (72%, 95% CI 64–79%) and kurtosis (62%, 95% CI 54–70%) as shown in Figure 3. All four features were tested for multicollinearity using the Spearman correlation coefficient [25] as illustrated in Figure 4. While the correlation of the first three features (correlation, IMC1, IMC2) was moderate to strong, kurtosis showed almost no correlation with the remaining three features.

Based on the results, a multivariate predictive model was constructed combining these four features. Given the PRE-CT segmentations, the model classified each lesion as either “HCC-probable” or “HCC improbable”. 60/82 (73%) lesions with later detected HCC were correctly classified as “HCC probable”, while 20/104 (19%) control regions were classified false positively as “HCC probable”. A total of 84/104 (81%) control-regions without HCC were classified correctly as “HCC improbable”, while 22/82 (27%) segmentations of later confirmed HCCs were not detected by the model and falsely classified as “HCC improbable”. The multivariate predictive model exhibited an AUC of 81% (95% CI 74–87%), a sensitivity of 72% (95% CI 57–83%), and a specificity of 86% (95% CI 76–96%), as illustrated in Figure 5. When including alpha-fetoprotein (AFP) levels at the time of the PRE-CT as a non-imaging, fifth input into the multivariate logistic regression model, AUC did not change significantly but rather declined slightly (79%, 95% CI 72–86%) with a sensitivity 71% (95% CI 56–83%) and a specificity 85% (95% CI 74–95%). AFP levels at the time of PRE-CT were available in 44 of 51 (86%) cirrhotic patients with newly emerged HCC and in 15 of 22 (68%) cirrhotic patients without HCC-recurrence. The detailed results of the multivariable analysis with and without AFP are available as Appendix A.

## 4. Discussion

To improve clinical care, early detection of HCC in high-risk patients with liver cirrhosis is paramount to prevent descent into the BCLC tumor stage. A timely detection could potentially avoid palliative systemic therapy and might render local tumor therapy an option instead. The aim of this study was therefore to investigate the potential benefit of an artificial intelligence prediction model for the assessment of CECT during surveillance of HCC patients. This was achieved by the identification of independent radiomic features and subsequent integration in a multivariate prediction model.

The results of this study prove that a multivariate model using four distinct radiomic features can predict the de-novo development of HCC based on CECT scans in AP in a defined cirrhotic region of the liver, with a quite acceptable AUC of 81% (95% CI 74–87%) and corresponding levels of sensitivity and specificity of 72% (95% CI: 57–83%) and 86% (95% CI: 76–96%), respectively. In comparison, a reader study by Seeman et al. found performance levels for the detection of HCC in CECT of human readers (radiologists) to be slightly inferior with regards to AUC (70%, 95% CI 66–74%) and sensitivity (60%, 95% CI 54–65%). Specificity was yet notably higher with 97% (95% CI 93–99%) [13]. Of course, the determined results of the two studies are not fully translatable and the comparison should be taken with caution. Still, considering that the segmented areas used for prediction of HCCs (PRE-CT) were not yet detected by radiologists in clinical practice, the results still seem promising: In the study cohort, the time of diagnosis could in theory have been shortened by about four months (median time interval between PRE-CT and FU-CT: 121 days). Based on these findings, the supportive use of radiomic analysis in the assessment of CECTs of high-risk patients should be considered and further investigated to potentially increase the detection rate and shorten the time of diagnosis and therefore yield potentially more beneficial treatment options for patients.

Similar, yet notably slightly inferior results have been found in a similarly designed study, in which indeterminate liver nodules were classified between HCC and no HCC using quantitative imaging features extracted from triphasic CT scans [26]. The authors identified a single radiomic feature and hereby reached an AUC of 70%. Another study examined the use of radiomic analysis for the recurrence prediction for HCC after liver transplantation in CECT [27]. Based on a combined model consisting of the radiomics signature and clinical risk factors, the authors were able to achieve a comparable predictive performance of 79%. Moreover, their prediction model was also solely based on the AP, which showed a better performance than the portal vein phase (PVP) or the fusion signature combining both phases, AP and PVP. This is one reason that prompted this study to base the model only on radiomics features extracted from AP. The other reason is simplicity. Since the predictive model requires the reporter to manually draw a segmentation slice-by-slice around an area of concern, the reporter would have to perform this time-consuming chore twice. If the patient has moved between the scans in AP and PVP or taken a different breathing position, the result might even be falsified due to the non-availability of anatomical matching, thereby rendering the process error prone. One could argue that without the images in PVP, the additional possibly relevant information is missing. Although this is certainly true, PVP is also associated with larger lesion size [28] and therefore potentially rather negligible in the prediction or early detection of initially smaller HCC nodules. Certainly, this matter and a potential benefit of PVP for a predictive model needs to be investigated in further studies.

The AFP level is known to improve the screening-based detection of HCC [29] and the combination of image-based radiomic features with clinical parameters such as Child-Pugh score, tumor size or AFP level has already been established to improve the prediction of survival [26,27,30]. Thus, further enhancement of the predictive power of the model was investigated by the addition of the AFP value at the time of PRE-CT as a non-imaging parameter. However, the results with an AUC of 79% (95% CI 72–86%), a sensitivity of 71% (95% CI 56–83%), and a specificity of 85% (95% CI 74–95%) showed no superiority over the purely image-based multivariate model. This could be because the future HCC nodule is yet too small to produce a significant increase in AFP. In addition, it should be noted that the AFP level was available in only 44 of 51 cirrhotic patients with HCC and in 15 of 22 cirrhotic patients without HCC, limiting the patient cohort. Furthermore, some patients already had pre-existing, sometimes even larger HCC lesions, which represents a potential significant confounder. A larger sample of patients without the presence of additional HCC-manifestations would be required to evaluate the predictive performance of AFP levels in a multivariate radiomic analysis of HCC in CECT.

While the results of this proof-of-concept study are promising, there are relevant limitations that need to be addressed: A major drawback of the predictive model is the fact that it is currently still limited to manually selected regions of cirrhotic liver tissue. The limited number of patients did not allow a sufficient analysis of the whole liver tissue at once. To date, the reporter needs to select one or more regions of concern where he believes HCC is likely to develop. The overall prediction statement, therefore, rests on the reporter’s choice. To be widely used in routine surveillance, the predictive model needs further development to automatically assess the entire cirrhotic liver. Another limitation is the relatively low number of patients, which is a general issue of developing predictive models in medical science. However, considering the deliberately strict inclusion criteria, the total number of 186 lesions (82 HCCs and 104 cirrhotic liver areas without HCC) can be considered quite reasonable. In contrast, the major benefit of this study is that it exclusively included cases in which the emergence of new HCCs could be investigated and even validated in the same individual over time. In addition, all examinations were performed on uniform CT scanners with standardized examination protocols, so that the model is rather robust against technical inhomogeneities. However, to achieve broader applicability, future work must also focus on external validation of using datasets acquired at different centers.

## 5. Conclusions

As applied in this proof-of-concept study, radiomic features extracted from arterial-phase CT imaging facilitate the prediction of newly emerging HCCs within segmented areas of cirrhotic livers with acceptable sensitivity and specificity. This could potentially shorten the time to detection of HCC and thus aid the diagnostic process of high-risk patients. Though, the procedure is currently limited by its restriction to preselected regions of interest in the liver, which hinders broader applicability. Clinical validation, as well as comparison to MRI (gold standard), is warranted. Further studies with larger patient numbers could potentially overcome this limitation.

## Figures and Tables

**Figure 1 diagnostics-11-01650-f001:**
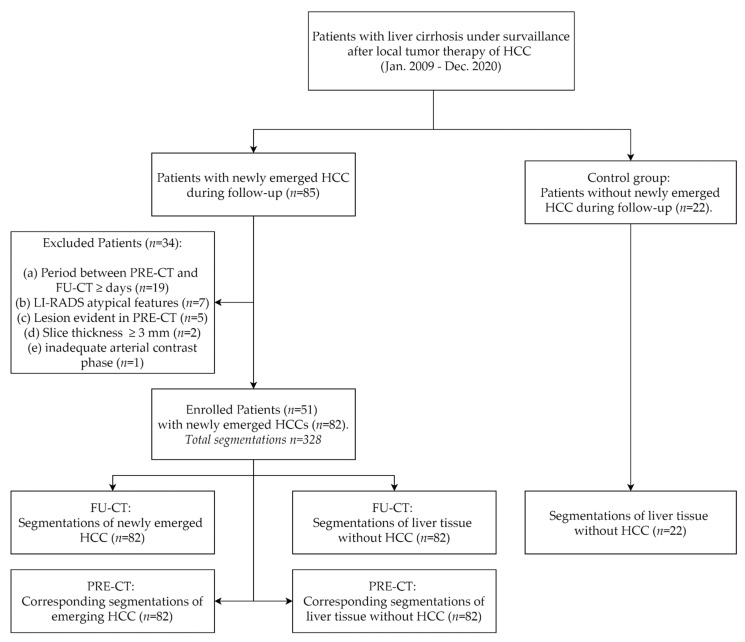
Flow chart of enrolled patients in the study and segmentations derived therefrom. HCC—hepatocellular carcinoma; PRE-CT—computed tomography prior to detection of a newly emerged HCC, FU-CT—follow-up computed tomography at the time of detection of a newly emerged HCC.

**Figure 2 diagnostics-11-01650-f002:**
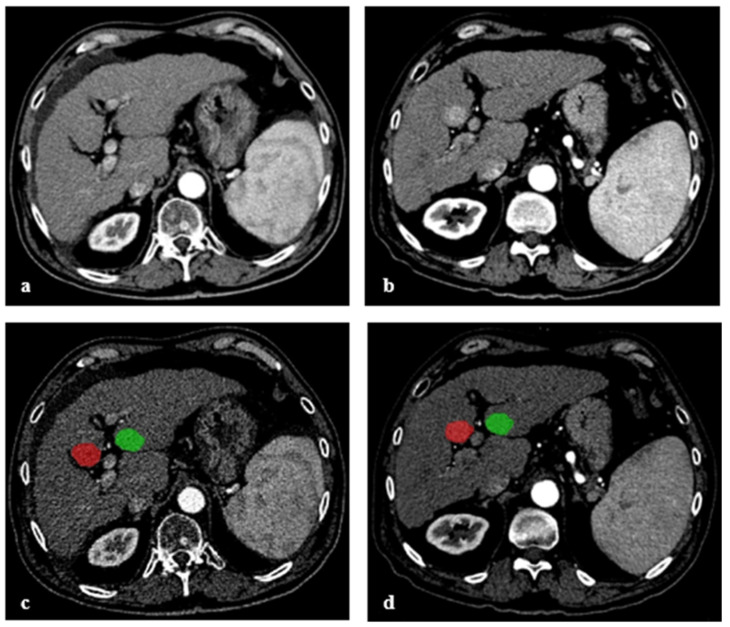
Arterial phase computed tomography of a cirrhotic patient with no visually detectable lesion on initial CT (PRE-CT) (**a**). CT 6 months later (FU-CT) shows a newly emerged HCC lesion in liver segment V (**b**). Segmentations used for the analysis in arterial phase images of PRE-CT (**c**) and FU-CT (**d**). Segmentations were first drawn around the visible border of HCC in FU-CT and subsequently in the corresponding region in PRE-CT, shown in red. As a negative control, a segmentation comparable in size and shape was drawn in PRE-CT and FU-CT in a region without manifestation of HCC, shown in green.

**Figure 3 diagnostics-11-01650-f003:**
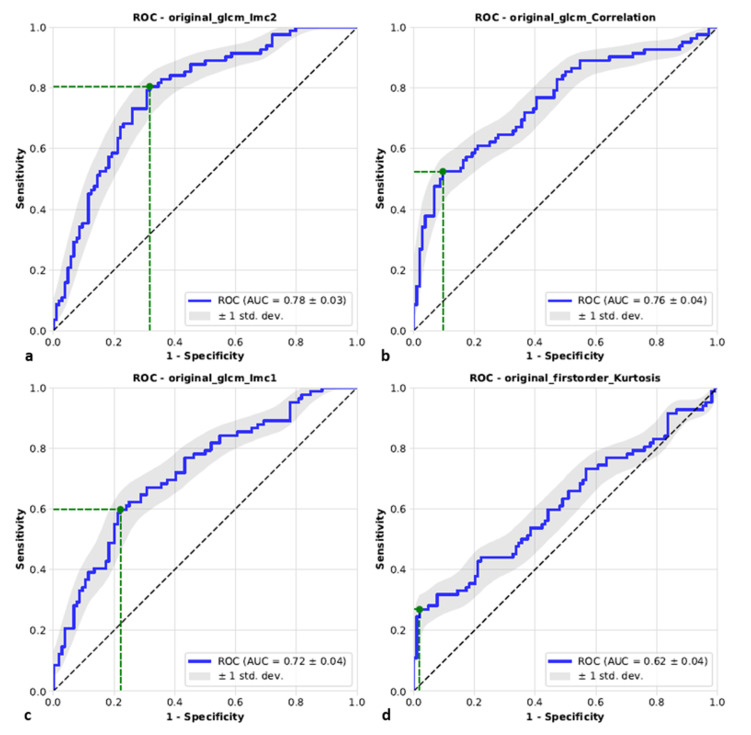
Receiver operating characteristics curves (ROCs) demonstrating the accuracy of each of the four univariate predictive models in the binary prediction for informational measure of correlation 2 (IMC2) (**a**), correlation (**b**), informational measure of correlation 1 (IMC1) (**c**) and kurtosis (**d**). The ROCs are based on 186 segmentations in PRE-CT, thereof 82 segmentations with HCC later confirmed in the FU-CT and 104 segmentations without HCC in the FU-CT. The working point was determined by maximizing sensitivity + specificity. IMC1 and IMC2 and correlation represent texture features, whereas kurtosis is a representative of the first-order features and specifies the peaked-ness of the distribution of values in the segmentation. GLCM—gray level co-occurrence matrix features, HCC—hepatocellular carcinoma, PRE-CT—computed tomography prior to detection of a newly emerged HCC, FU-CT—follow-up computed tomography at the time of detection of a newly emerged HCC.

**Figure 4 diagnostics-11-01650-f004:**
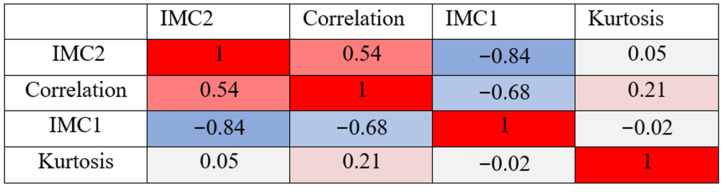
The Spearman rank-order correlation coefficient between the four radiomics features used in the multivariate predictive model. It varies between −1 and +1 with 0 implying no correlation. Correlations of −1 or +1 imply an exact monotonic relationship. Positive correlations imply that as one feature increases, so does the other feature, whereas negative correlations imply that as one feature increases, the other decreases. White—no correlation, Red—positive correlation, Blue—negative correlation.

**Figure 5 diagnostics-11-01650-f005:**
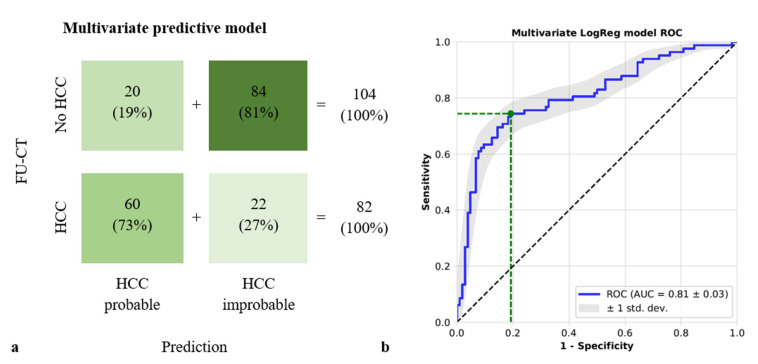
(**a**) Confusion matrix for the prediction of HCC in segmentations of the cirrhotic liver in the multivariate model integrating the four features (correlation, IMC1, IMC2, and kurtosis). IMC1 and IMC2 (informational measure of correlation 1 and 2) and correlation represent texture features, whereas kurtosis is a representative of the first-order features. (**b**) ROC for the multivariate model including the four imaging features outlined above for dichotomizing segmentations into HCC probable and HCC improbable regions. HCC—hepatocellular carcinoma.

**Table 1 diagnostics-11-01650-t001:** Epidemiologic and clinical characteristics. IQR—interquartile range, MTD—median tumor diameter, HCC—hepatocellular carcinoma, PRE-CT—computed tomography prior to detection of a newly emerged HCC, FU-CT—follow-up computed tomography at the time of detection of a newly emerged HCC.

Epidemiologic and Clinical Characteristics	Cirrhotic Patients with HCC Lesions in the FU-CT	Cirrhotic Patients without HCC Lesions under Surveillance
Study size	51	22
Age (years), median (IQR)	69 (63–76)	55 (51–63)
Gender, no. (%)		
Male	39 (76)	14 (64)
Female	12 (24)	8 (36)
Total number of segmentations with HCC/without HCC	82/82	0/22
MTD (cm) in FU-CT, median (IQR)	1.2 (0.9–1.5)	
Tumor location in FU-CT, no. (%)		
Left lobe	41 (50)	
Right lobe	41 (50)	
Time interval between PRE-CT and FU-CT (days), median (IQR)	121 (94–165)	

**Table 2 diagnostics-11-01650-t002:** Predictive scores with 95% confidence interval for all univariate models and the multivariate model with all four radiomic features (correlation, IMC1, IMC2, kurtosis). IMC1 and IMC2 (informational measure of correlation 1 and 2) and correlation represent texture features, whereas kurtosis is a representative of the first-order features. AUC—area under curve, IMC—informational measure of correlation. *p*-values were calculated using bootstrapping.

Feature	Univariate				Multivariate
	Correlation	IMC1	IMC2	Kurtosis	All 4 Features
*p*-value	<0.000000001	<0.00000021	<0.000000001	<0.00007	
Sensitivity (%)	57 (44–90)	65 (51–86)	81 (65–90)	33 (21–79)	72 (57–83)
Specificity (%)	89 (51–97)	76 (52–86)	71 (59–83)	96 (47–100)	86 (76–96)
Threshold	0.22 (0.15–0.24)	0.04 (0.03–0.05)	0.29 (0.28–0.33)	3.36 (2.97–3.71)	0.45 (0.39–0.61)
AUC (%)	76 (68–83)	72 (64–79)	78 (72–85)	62 (54–70)	81 (74–87)
Weighting in multi-variate model (%)	23.5	26.2	35.2	15.1	

## Data Availability

The data that support the findings of this study are available from the corresponding authors upon reasonable request. The data are not publicly available due to it contain patient information.

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
