# Peer review of "A Radiomics Approach to Predict the Emergence of New Hepatocellular Carcinoma in Computed Tomography for High-Risk Patients with Liver Cirrhosis"

_diagnostics, 2021, doi:10.3390/diagnostics11091650_

Round 1

Reviewer 1 Report

This investigation was performed in order to identify radiomic features based on contrast-enhanced computed tomography and to establish an imaging procedure to predict the appearance of de novo HCC nodules in cirrhotic patients. The investigation was well conducted and is of interest for improving the early diagnosis of HCC; however, some issues still need to be addressed.

Based on statistical values authors conclude that their imaging procedure had an acceptable sensitivity and specificity. In methods section they also state that the radiologist was an expert in the field. So, this conclusion implies that the experience of the radiologist had a significant contribution in order to get a novel and acceptable procedure. Thus, a clear limitation that was not mentioned/discussed in the investigation is about the radiologist expertise, which represent a key factor both for performing an accurate abdominal imaging screening in patients bearing cirrhotic liver with high-risk of developing HCC and for reaching an acceptable procedure. Authors should discuss the key role played by an expert on the field in order to get the more accurate diagnosis.

This manuscript contains some syntax issues that need to be carefully reviewed and corrected.

Abbreviations in the title should be avoided.

Author Response

“Based on statistical values authors conclude that their imaging procedure had an acceptable sensitivity and specificity. In methods section they also state that the radiologist was an expert in the field. So, this conclusion implies that the experience of the radiologist had a significant contribution in order to get a novel and acceptable procedure. Thus, a clear limitation that was not mentioned/discussed in the investigation is about the radiologist expertise, which represent a key factor both for performing an accurate abdominal imaging screening in patients bearing cirrhotic liver with high-risk of developing HCC and for reaching an acceptable procedure. Authors should discuss the key role played by an expert on the field in order to get the more accurate diagnosis.”

We thank the reviewer for this insightful comment and would like to explain this point in more detail in the following: All cases of patients in follow-up for HCC, who were treated in our department (local ablation or transarterial tumor embolization) in the period of January 2009 to January 2021, were retrospectively reviewed by an experienced radiologist. Only cases with newly emerged HCC lesions during follow-up were selected for the study. According to the LI-RADS criteria a new HCC was defined by arterial enhancement and late venous washout of a newly depictable lesion with a size >1 cm. If the lesion was <1 cm, it was only included if clearly verifiable as HCC in subsequent follow-up. Since we applied highly standardized inclusion criteria, we consider the radiologist's experience to be quality-assuring but not strictly indispensable in this context. We have attempted to present the process of case selection by the radiologist in the methodology section somewhat more clearly now.

“This manuscript contains some syntax issues that need to be carefully reviewed and corrected.”

We thank the reviewer for this information and have carefully checked the manuscript for syntax errors and have made corrections in this regard.

“Abbreviations in the title should be avoided.”

This is indeed an important note, we have removed all abbreviations from the title.

Reviewer 2 Report

Interesting study evaluating radiomics of HCC.

The study is well written.

The number of patients remains low however the results are promising and aims original.

It would be interesting to have all the results of multivariate analysis in a table add as supplementary materials.

A validation cohort would be also helpful to assess the accuracy of parameters.

Please avoid the use of first person (we, our).

I would suggest to down the tone of the conclusion: radiomics would help... as a validation is still required as well a comparison to MR results which remains the gold standard in liver exploration.

Author Response

“It would be interesting to have all the results of multivariate analysis in a table add as supplementary materials.”

We thank the reviewer for the interest in the results and have added an appendix with the results of the multivariate analysis.

“A validation cohort would be also helpful to assess the accuracy of parameters.”

We fully agree with the reviewer that validation cohorts would increase the value of the work. We used an internal validation cohort and took great care in isolating the validation from the training. The treatment and detailed follow-up of large number of HCC patients is usually reserved to clinical centers, therefore no public dataset that we could have used for external validation was available. Due to data privacy regulations access of data from other University hospitals is currently only hardly possible. Even though our study is a proof of concept, we agree that external validation should be done in the future, and we added a sentence stating this in the discussion section for future work.

“Please avoid the use of first person (we, our).”

We thank the reviewer for this advice and have removed any first person references from the manuscript.

“I would suggest to down the tone of the conclusion: radiomics would help... as a validation is still required as well a comparison to MR results which remains the gold standard in liver exploration.”

We fully agree with the reviewer and have downgraded the weight of the conclusion. We also have added the need for clinical validation and comparison to MRI as the gold standard.

Reviewer 3 Report

  • The total segmentations for analysis were 168 that included 82 HCC FU-CT, 82 non-HCC FU-CT and 22 negative control. How were the 82 segmentations for HCC PRE-CT applied in the analysis?
  • Please explain briefly how the radiomic features in the segmentations from HCC FU-CT, non-HCC FU-CT, HCC PRE-CT and negative control groups were analyzed.
  • In this study, the predictive model used Pre-CT and FU-CT for comparison, please explain how this model can predict the occurrence of HCC better than the visual interpretation of the images to help to increase the detection rate and to shorten the time of diagnosis.
  • How did the places of segmentations of the 22 HCC free patients for the negative control determined?
  • P9 L249 : “BL CT” what is the abbreviation for BL?
  • P2 L81~83: Several newly detected lesions did not yet meet the formal criteria of definite HCC “several scans” due to a size <1cm.-->I don’t quite catch the meaning.

Author Response

“The total segmentations for analysis were 168 that included 82 HCC FU-CT, 82 non-HCC FU-CT and 22 negative control. How were the 82 segmentations for HCC PRE-CT applied in the analysis?”

We thank the reviewer for pointing out the difference between segmentations with and without HCC in PRE-CT,  FU-CT, and segmentations and in the control group. 51 patients with a total of 82 newly emerged HCC lesions were included in the study. For each HCC lesion, a control area of tumor-free liver tissue was segmented in the same individual, resulting in a total of 82*2=164 lesions. These lesions were then segmented on both PRE-CT and FU-CT, resulting in 82 * 4 = 328 segmented regions. The control group consisted of 22 randomly selected patients without HCC recurrence, here 22 tumor-free liver areas were segmented. Thus, a total of 328 + 22 = 350 segmentations were available for the entire analysis. Since this may not have been emphasized clearly enough in the manuscript, we have revised our flow-chart of the study protocol (Figure 1) and have specified this more precisely in the results.

“Please explain briefly how the radiomic features in the segmentations from HCC FU-CT, non-HCC FU-CT, HCC PRE-CT and negative control groups were analyzed.”

In the training process, a set of radiomic features was extracted from HCC PRE-CT and HCC FU-CT and compared to an identical set of radiomic features from non-HCC PRE-CT and non-HCC FU-CT, thereby applying a patient-by-patient leave-one-out cross-validation. That is, multiple models were trained by leaving one patient out of the training set while training with the remaining set of patients. Using univariate analysis, we identified four features after Bonferroni correction that allowed differentiation of cirrhotic liver tissue with and without subsequent HCC. Three of them (correlation, informational measure of correlation 1 and informational measure of correlation 2) were texture features (gray level co-occurrence matrix features = GLCM). The fourth feature (kurtosis) was a first order feature that specifies the peakedness of the distribution of values within the segmentation. Employing these four radiomic features, a multivariate logistic regression model was subsequently used to predict the development of HCC in each segmentation of PRE-CTs (HCC and Non-HCC).

“In this study, the predictive model used Pre-CT and FU-CT for comparison, please explain how this model can predict the occurrence of HCC better than the visual interpretation of the images to help to increase the detection rate and to shorten the time of diagnosis.”

We thank the reviewer for this quite important question. As stated above, only PRE-CT-images were used in our radiomics-approach for the prediction of HCC. Thus, our prediction model was able to predict the development of HCC based only on the outlined region in PRE-CT. This was on average four months before a clinical reader was able to identify the new lesion in the FU-CT. As described in the previous response, the radiomic features used in the model were texture and kurtosis parameters, which are directly affected by the density of the segmented tissue. 

We also questioned why these lesions could be generally predicted by the model, but not by visual assessment by the radiologist. In our opinion, this was due to the fact, that although some of the lesions appeared to be slightly increased in density on PRE-CT, they were not objectively classifiable as an arterial enhancement typical for HCC.

In addition, a relevant shortcoming of our model must be considered in this context:
The prediction of HCCs is only possible in segmented liver regions with the mentioned diagnostic metrics. These segmentations were precisely drawn by the radiologist in such a way that regions of the developing HCCs overlap exactly in PRE-CT and FU-CT. Of course, the transferability to clinical routine is limited as a result. Thus, we do not conclude that our model is superior to a radiologist, but rather that in this context, prediction of the developing HCC is possible with moderate accuracy.

“How did the places of segmentations of the 22 HCC free patients for the negative control determined?”

The locations of the negative control lesions were chosen randomly by the radiologist in vascular free areas of the cirrhotic liver tissue, with respect to a comparable distribution over all liver lobes and comparable size to HCC-segmentations.

“P9 L249 : “BL CT” what is the abbreviation for BL?”

We apologize to the reviewer for this typographical error, it was a formerly used abbreviation for PRE-CT. The error has been corrected accordingly.

“P2 L81~83: Several newly detected lesions did not yet meet the formal criteria of definite HCC “several scans” due to a size <1cm.-->I don’t quite catch the meaning.”

This is indeed an important point, which we are happy to explain in more detail: According to the LI-RADS criteria, lesions were classified as HCC if they showed an arterial enhancement and late venous washout and a lesion size >1 cm. However, a few newly detected lesions did not yet meet the formal criteria of definite HCC due to a lesion size  below 1cm. As these lesions were obviously HCCs, despite its size below 1 cm, they were included if verifiable as HCCs based on further tumor progression in subsequent follow-up CTs. We have revised this section to provide a better understanding.

Reviewer 4 Report

Reviewer comments and suggestions

The current research was a retrospective approach to investigate the extended radiomic features that allow the prediction of emerging HCC in patients with liver cirrhosis in contrast-enhanced computer tomographies (CECT). 51 patients with a history of HCC and liver cirrhosis as well as the emergence of a new HCC during follow-up examinations (FU-CT) were included. A total of 186 segmentations (82 HCCs and 104 cirrhotic liver areas without HCC) were analyzed. The author used multivariate logistic regression model was to classify the outlined regions as “HCC probable” or “HCC improbable”. The results indicated that the model used in the study predicted the occurrence of new HCCs within segmented areas with the help of the AOC curve. 

Decision: Minor comments

Below are the comments for this paper to be incorporated in the revised version of the manuscript. 

  1. Line 20-21 The sentence need to be modified “51 patients with history of HCC and liver cirrhosis as well as emergence of a new HCC 20 during follow-up examinations (FU-CT) were included”
  2. Line 28 HCC improbable ? not included 
  3. Line 38 what does it mean? resulting in overall lifetime risk of roughly 30% [4]
  4. Line 54, first time used MRI, Full form needed
  5. Line 64-65 needs to modify. “However, no study has utilized 64 radiomics to not only detect but to predict new development of HCC in patients with liver 65 cirrhosis using CECT imaging”
  6. Line 78-79 how this would be follow up CT
  7. Section 2.5. Development of a prediction model it needs to be discussed briefly
  8. Figure 4 2.5. Development of a prediction model
  9. Line 203 was not discussed previously
  10. Line 209-211 needed to add information about multivariate in the legend of the figure
  11. Line 217-220 no need to repeat the lines as they can be useful in the introduction
  12. Line 224-226 The author needed to discuss the result not only writing the specific result
  13. Line 277-278 is this conclusion para, please recheck it
  14. The author can modify the conclusion by using lines 274-285
  15. Almost all references need to be modified based on the MDPI journal format.

Author Response

  1. “Line 20-21 The sentence need to be modified “51 patients with history of HCC and liver cirrhosis as well as emergence of a new HCC 20 during follow-up examinations (FU-CT) were included”

We thank the reviewer for the comment and have now chosen a more understandable wording.

  1. “Line 28 HCC improbable ? not included “

We thank the reviewer for this important reference and have now provided the results for both classifications "HCC probable" and "HCC improbable" in the abstract.

  1. “Line 38 what does it mean? resulting in overall lifetime risk of roughly 30% [4]”

The sentence should state that in the presence of cirrhosis there is a risk of approximately 30% of developing HCC during one's lifetime. Since the sentence may have been worded unclearly in the present version, we have now attempted to phrase it more precisely.

  1. “Line 54, first time used MRI, Full form needed”

We thank the reviewer for pointing this out and made appropriate adjustments.

  1. “Line 64-65 needs to modify. “However, no study has utilized 64 radiomics to not only detect but to predict new development of HCC in patients with liver 65 cirrhosis using CECT imaging”

We thank the reviewer for pointing this out and have now enhanced the comprehensibility of the mentioned statement.

  1. “Line 78-79 how this would be follow up CT”

We thank the reviewer for the comment and understand that the comprehensibility of this section was not yet sufficient. In conjunction with the additional suggestions from Reviewer 1, we have thoroughly revised this section and hope that it is now more concise and comprehensive.

  1. “Section 2.5. Development of a prediction model it needs to be discussed briefly”

We fully agree with the reviewer and added a respective discussion of the development of the prediction model.

  1. “Figure 4 2.5. Development of a prediction model”

We apologize for the lack in clarity and thank the reviewer for the important reference. To explain the prediction model in more detail to the attentive reader we have added a description of the process of feature selection and test for multicollinearity following paragraphs to 2.5: “Each feature was adjusted for mean and standard deviation to achieve uniform scaling between features.  Based on the estimated HCC probabilities and the actual occurrence of HCC, the optimal threshold in terms of maximum sensitivity and specificity was determined in a ROC curve. This threshold was then used to classify PRE-CT images into "HCC probable" and "HCC improbable". Features were tested for multicollinearity using Spearman correlation coefficient.“ In addition, we have added more detailed descriptions to the legends of Figure 3 and Figure 4 to enhance comprehensibility.

  1. “Line 203 was not discussed previously”

We thank the author for the note. The experimental integration of the AFP value, which was only briefly described in the section "Development of a prediction model", is now explained in more detail in the results and discussion.

  1. “Line 209-211 needed to add information about multivariate in the legend of the figure”

We thank the reviewer for this important reference. We added Information regarding the four radiomic features used in the multivariate model in the legend.

  1. “Line 217-220 no need to repeat the lines as they can be useful in the introduction”

This is of course true, the repetitive statement in the results was omitted.

  1. “Line 224-226 The author needed to discuss the result not only writing the specific result”

We thank the reviewer for pointing this out. The paragraph in question has been supplemented by comparisons with other studies and we further discussed the potential advantages of the applied approach.

  1. “Line 277-278 is this conclusion para, please recheck it”

We apologize for this oversight and have revised the conclusion accordingly.

  1. “The author can modify the conclusion by using lines 274-285”

Again, we thank the reviewer for pointing this out, revisions were made in conjunction with point 13.

  1. “Almost all references need to be modified based on the MDPI journal format.”

We thank the reviewer for pointing this out. The reference section was modified based on the MDPI journal format.

Round 2

Reviewer 2 Report

Thank you for your review.